# Cartilage Homeostasis Affects Femoral Head Necrosis Induced by Methylprednisolone in Broilers

**DOI:** 10.3390/ijms21144841

**Published:** 2020-07-08

**Authors:** Yaling Yu, Shujie Wang, Zhenlei Zhou

**Affiliations:** Department of Veterinary Clinical Science, College of Veterinary Medicine, Nanjing Agricultural University, Nanjing 210095, China; 2018107108@njau.edu.cn (Y.Y.); 2018207034@njau.edu.cn (S.W.)

**Keywords:** cartilage homeostasis, femoral head necrosis, methylprednisolone, hypoxia-inducible factors, broiler

## Abstract

(1) Background: Since the large-scale poultry industry has been established, femoral head necrosis (FHN) has always been a major leg disease in fast-growing broilers worldwide. Previous research suggested that cartilage homeostasis could be taken into consideration in the cause of FHN, but the evidence is insufficient. (2) Methods: One-day-old broiler chickens were randomly divided into three groups, 16 broilers per group. The birds in group L were injected intramuscularly with methylprednisolone (MP) twice a week for four weeks (12.5 mg·kg^−1^). The birds in group H were injected intramuscularly with MP (20 mg·kg^−1^·d^−1^) for 7 d (impulse treatment). The birds in group C were treated with sterile saline as a control group. Broilers were sacrificed at 42 and 56 d. Blood samples were collected from the jugular vein for ELISA and biochemical analysis. Bone samples, including femur, tibia, and humerus, were collected for histopathological analysis, bone parameters detection, and real-time quantitative PCR detection. (3) Results: The FHN broilers in group L and H both showed lower body weight (BW) and reduced bone parameters. In addition, the MP treatment resulted in reduced extracellular matrix (ECM) anabolism and enhanced ECM catabolism. Meanwhile, the autophagy and apoptosis of chondrocytes were enhanced, which led to the destruction of cartilage homeostasis. Moreover, the impulse MP injection increased the portion of birds with severer FHN, whereas the MP injection over a long period caused a more evident change in serum cytokine concentrations and bone metabolism indicators. (4) Conclusions: The imbalance of cartilage homeostasis may play a critical role in the development of FHN in broilers. FHN broilers induced by MP showed a more pronounced production of catabolic factors and suppressed the anabolic factors, which might activate the genes of the WNT signal pathway and hypoxia-inducible factors (HIFs), and then upregulate the transcription expression of ECM to restore homeostasis.

## 1. Introduction

Over the past decades, the commercial poultry industry has a significant rapid due to the application of genetics, nutrition, and proper management, the most remarkable changes being feed conversion and body weight (BW) [1]. The type of broiler commonly used today grows 300% faster than those in the 1960s [2]. However, significant weight gain causes many problems, such as pulmonary hypertension, fatty liver syndrome, and an increasing incidence of skeletal problems in young poultry [3,4,5,6]. Broilers gain the most weight between four and six weeks of age. The sudden pressure on the legs leads to various leg problems [7]. Clinical and morphological investigations of broilers had revealed that the incidence rate of lameness varied from 3–4% up to 15% in farms [8]. Moreover, many large-scale field studies demonstrated that femoral head necrosis (FHN) was the most common cause of lameness in broilers [2,8,9,10]. Most studies suggested that the causes of FHN may be related to an insufficient blood supply to the femoral head [11,12,13] or disorders of lipid metabolism [14,15,16,17], but the mechanism of FHN is rather complicated and not completely clear. Previous research has demonstrated that the pathogenesis of methylprednisolone (MP)-induced FHN is similar to naturally occurring FHN, which made MP the major method of establishing the FHN model [18,19,20].

FHN is associated with cartilage damage [21]. It is well known that the articular cartilage mainly consists of chondrocytes and extracellular matrix (ECM). In addition, chondrocytes are the only cells in the articular cartilage that synthesize the whole ECM, including proteoglycan, glycosaminoglycan, and type II collagen [22,23,24]. Under normal conditions, chondrocytes continuously synthesize new matrices and degrade the aging matrix to maintain the balance of the articular cartilage, and this process is called cartilage homeostasis [25,26,27]. However, chondrocytes are few in number and have a low metabolic dynamic. Therefore, the dynamic equilibrium between the synthesis (anabolism) and degradation (catabolism) of ECM components is not only based on a direct regulation of chondrocytes but also on various factors secreted by them and other cells [22,23,28]. The factors that could maintain the homeostasis of ECM are divided into catabolic cytokines, including interleukin-1β (IL-1β), tumor necrosis factor α (TNF-α), and anabolic factors, e.g., cartilage morphogenetic protein (CDMP), bone morphogenetic protein (BMP), and insulin-like growth factor 1 (IGF-1) [23,29,30,31].

Chondrocyte homeostasis is coordinated through a series of physiological processes, such as apoptosis, autophagy, and cell proliferation [21,32]. Our previous studies have proved that the apoptosis of chondrocytes plays a vital role in the development of FHN [5,21], and autophagy has also been reported to be involved in the development of osteoarthritis (OA) [33]. As the abnormal autophagy and apoptosis can disturb homeostasis of cartilage chondrocytes and cause cartilage damage, the regulatory mechanism of cartilage homeostasis should be taken seriously in the initiation and pathogenesis of FHN.

The articular cartilage is devoid of blood vessels, lymphatics, and nerves, which makes chondrocytes sensitive to the changes of the extracellular environment. The perception and response to oxygen is a critical aspect of cartilage homeostasis, it is mainly adjusted by hypoxia-inducible factors (HIFs) and its regulator von Hippel–Lindau tumor suppressor protein (pVHL). HIFs can also induce the expression of proteins which controls glucose metabolism, cell proliferation, and angiogenesis. Furthermore, it is also a key regulator that regulates articular cartilage homeostasis and chondrocyte activities [22,32,34,35,36], but there is little research into the regulatory mechanisms of homeostasis. Recent studies have shown that the function of HIF-1α is regulated by pVHL, which might be associated with the WNT signaling pathway [37,38]. Therefore, HIFs and WNT signaling proteins might play a crucial part in regulating craticular homeostasis in the development of FHN.

Despite extensive studies, the fundamental mechanisms responsible for the relationship between FHN and the homeostasis of the articular cartilage in broilers and other animals have not yet been fully elucidated. The aim of this study is to investigate how cartilage homeostasis affects FHN induced by MP in broilers. The expression of anabolism relation genes (CDMP-1, BMP-3, VEGFA, and IGF-1) and catabolism relation genes (MMP-9, IL-1β), and the development of apoptosis and autophagy in the articular cartilage were detected. Bone metabolism indicants and bone parameters were examined. In addition, the levels of serum biochemical indicators, including creatinine, alkaline phosphatase (ALP), calcium (Ca), and phosphorus (P) were also tested.

## 2. Results

### 2.1. Morbidity of FHN and Feed Conversion

The experiment was finished at 56 d, two birds in each MP-treated group died before 42 d. Therefore, the results of 42 d were obtained from 18 chickens (six chickens per group) and the results of 56 d were obtained from 24 chickens (eight chickens per group). Behavioral observations (see Table 1) showed that the broilers in group H had more serious gait defect than those in group L, and there was only one 56-d broiler in group H that had a gait score of 5 (the bird was incapable of sustained walking on its feet). In addition, broilers in group H had diarrhea during continuous administration, and the symptom gradually disappeared after the administration. For the same bird, different legs had different symptoms; therefore, the morbidity of broiler FHN was calculated by the total amount of legs. The evaluation and classification of legs were based on the standards of Okazaki et al. [39]. The macroscopic morphological changes of FHN are shown in Figure 1, and results of FHN evaluation are listed in Table 2. In group H, which used the pulse MP treatment, there was an evident rise both in morbidity and the total FHN score compared with group C at the age of 42 d. The morbidity and total FHN score of group L increased significantly from 42 to 56 d.

Throughout the whole experiment, the speed of gaining weight in two MP-treated groups was always slower than group C (see Figure 2). Moreover, there was no significant difference in feed conversion among the three groups. The feed utilization had increased before the 35th day and then slowly decreased. Furthermore, there was no difference in the liver index in all groups (see Table 3).

### 2.2. Changes in ECM Homeostasis Factors

The changes of catabolic cytokines (IL-1β) are shown in Figure 3A. The level of catabolic cytokines in group H and L significantly increased compared with group C, and were higher in group L. The change trend in serum anabolic cytokines (IGF-1 and vascular endothelial growth factor, VEGF) of group L was in accordance with group H, especially at the age of 56 d (Figure 3B,C). These two factors, both in group L and H, were significantly decreased compared with group C, which were conversed with the IL-1β.

### 2.3. Bone Biochemistry and Bone Metabolism Indicants

The serum biochemical indicators are shown in Table 4. The creatinine had no significant difference between the three groups. The level of ALP in group H was higher than in group C. Meanwhile, after a long-term treatment with MP, the emission of Ca in the serum evidently increased.

Table 5 shows the changes of bone metabolism. The level of alkaline phosphatase (BALP) and osteocalcin (OT) in the two MP-treated groups significantly decreased compared with group C at the age of 56 d, indicating a lower level of bone formation indicants. Meanwhile, the two MP-treated groups showed that the bone resorption indicants, tartrate-resistant acid phosphatase 5b (TRACP-5b) and cross-linked carboxy-terminal telopeptide of type I collagen (CTX-I), were significantly increased, and group L had a significantly higher level than group H. In addition, compared with the 42-d indicants in group C, bone-formation indicants had increased and bone-resorption indicants had decreased at the age of 56 d.

### 2.4. Pathological Section and Bone Parameters

The hematoxylin and eosin (H&E) staining results of the femoral head articular cartilage of different groups are shown in Figure 4. Chondrocytes had a normal morphology and intact structure in group C, whereas there were more empty lacunae presented in MP-treated broilers, which meant that the structures of cartilage in group L and H were destroyed in various degrees.

According to the data in Table 6 and Table 7, the length, index, and diameter of chicken bone (humerus, femoral, and tibia) in the two MP-treated groups significantly decreased compared to group C at both 42 and 56 d. Meanwhile, the bone mineral density (BMD) in two MP-treated groups also decreased compared with group C, but there was no significant difference.

### 2.5. Expression of Homeostasis Relative Genes

qRT-PCR was used to evaluate the mRNA expression differences of target genes. Figure 5 shows the mRNA level of some factors related to ECM homeostasis of the articular cartilage. The 42-d group H showed that the expression of anabolism relation genes (CDMP-1, BMP-3, VEGFA, and IGF-1) were inhibited and the catabolism relation genes (matrix proteinase-9 (MMP-9), IL-1β) were increased. In group L, however, all seven factors had enhanced. Moreover, the expression of the four anabolism relation genes was significantly suppressed in group L and H at 56 d, and two catabolic factors were increased.

Figure 5 shows the change of cell homeostasis-related genes. Both group L and H at 42 d showed that the expressions of anti-apoptosis genes Bcl-2 and cell proliferation markers KI-67 and PCNA were slightly suppressed, whereas the autophagy relation ratio (LC3-II/LC3-I) and pro-apoptotic genes Bid were promoted. However, the expression of autophagy and apoptotic genes in group L did not have an evident change compared with group C at 56 d, whereas the pulse treatment showed a particular promotion of autophagy and apoptosis.

As for the factors of the WNT signaling pathway and HIFs, the expression of all these relative factors were promoted in group L and H at the age of 42 d. The 56-d group H showed a more evident increase compared to 42 d, whereas group L slightly inhibited most factors. The expressions of aggrecan, collagen-2, and collagen-10 visibly changed with the factors of the WNT signal pathway and HIFs.

## 3. Discussion

Glucocorticoid is one of the most common methods to induce FHN in various experimental animals [16,40]. FHN can be induced by various doses of glucocorticoid, different kinds of glucocorticoid, and diverse administration methods, yet the morbidity and mechanism remain unclear. Previous research suggested that there may be a relationship between articular cartilage homeostasis and FHN in broilers [20], but today there is still not enough evidence or convincing statistics. Meanwhile, the broiler is bipedal, meaning that it offers an inherent advantage for those studying the FHN of human beings [3,13,40,41,42]. This study used two different administration plans to build the models of FHN in broilers, with the aim to examine the difference in morbidity of FHN induced by a long-term and impulse MP injection, and explore the role of articular cartilage homeostasis in the pathogenesis of FHN.

MP has been widely used in a veterinary clinic for anti-inflammatory and immunosuppression, but there is little information available on birds [43]. Excessive use of MP can cause serious side effects, e.g., injecting dogs with 20 mg·kg^−1^ MP continuously for more than 25 days could produce fatal sequelae [44]. According to this study, both administration plans increased the morbidity of FHN in broilers, whereas the impulse MP injection caused more broilers to suffer from serious FHN with rupture of the femoral head, which might connect with lower BW and a higher transcription level of autophagy, apoptosis, WNT signal pathway, and HIFs in group H. Autophagy and apoptosis might play a vital role in the development of serious FHN.

Serum cytokines are the key to the regulation of ECM homeostasis [22,23,30]. Increased catabolic cytokines and reduced anabolic cytokines in both group L and H showed that MP could promote catabolism and inhibit anabolism to disrupt the balance of cytokines, which cause the destruction of cartilage ECM homeostasis and may then induce FHN. In addition, the long-term administration has a more evident change. Moreover, it has been reported that IL-1β is the crucial catabolic factor, which could inhibit the functions of growth factors, proliferation of chondrocyte, and the repairing of ECM. Meanwhile, it also induces apoptosis of chondrocyte and degradation of ECM [23]. Some research suggested that it could induce production of MMPs [27], which further accelerated ECM degradation [24,45]. The results showed that the serum concentration of IL-1β was increased at 42 and 56 d in both MP-treated groups. As for VEGF and IGF-1, their serum concentrations were significantly reduced in both MP-treated groups at 56 d. They have the opposite functions to catabolic factors [30,31,45]. In the present study, the change of catabolic cytokines was more significant, which had a negative relation with the bone-formation indicants and proliferation-related genes. Furthermore, catabolic factors also have a positive association with bone-resorption indicants and the transcription level of MMP-9. The catabolism of cartilage might be promoted in the early stage of FHN and through the whole FHN process, whereas the anabolism was inhibited in the later stage of FHN. In addition, the long-term MP treatment would make these changes more evident.

The dynamic balance of bone formation and resorption maintains the completeness of the skeleton. The bone metabolism indicants in the serum demonstrated that the bone formation was prevented while bone resorption was activated in two MP-treated groups. Furthermore, comparing the bone metabolism indicants of 42 and 56 d in group C showed that the bone was still in the growth phase, and treatment with the long-term MP had a more significant impact on the bone metabolism in broilers. The cytokines of the treated broilers have a close connection with these indicants, e.g., high-level catabolic factors might promote bone resorption and inhibit formation, meanwhile, anabolic factors which have converse functions were suppressed [46,47].

Bone quality is an important indicator of leg health, it can be evaluated by measuring bone parameters [48]. The significant reduction in BMD, bone index, and bone diameter indicated that MP inhibited the growth of bones and caused bone mineral loss [49]. In addition, the long-term MP treatment had a greater influence on bone quality than the impulse MP treatment, consistent with the results of bone metabolism indicants. Therefore, MP mainly deteriorates the bone quality by promoting bone resorption and reducing bone formation.

The transcription data indicated that FHN in broilers may be related to the homeostasis of ECM and cells. The results showed that anabolic factors (CDMP-1, BMP-3, VEGFA, and IGF-1) have declined after MP injection, which might play a major role in FHN [20]. However, group L showed that both catabolism and anabolism at 42 d are at a high level, and the reason for this phenomenon may be that the body is against the function of MP in the early stage of a long-term MP injection. In addition, the expression of IL-1β in group L increased dramatically. Huang et al. reported that IL-1β was a crucial factor in accelerating the degradation of cartilage ECM [50]. Therefore, as seen in the autopsy, the incidence of FHN in group L has greatly increased (from 25% to 43.75%) from 42 to 56 d.

It is well accepted that catabolic and anabolic factors could activate the chondrocyte [27]. Furthermore, chondrocytes have an orderly cell death and proliferation to stabilize the total number, while proliferation and apoptosis are the main factors involved in FHN [51]. After analyzing the 42-d factors related to cell renewal, the transcription level of autophagy and pro-apoptosis were promoted, whereas anti-apoptosis and proliferation were suppressed in MP-treated broilers. Therefore, it reveals that apoptosis and autophagy may promote the development of FHN. Additionally, the impulse MP-treated broilers had a higher expression of autophagy-related and pro-apoptosis genes, which could be considered as a reason for more severe FHN.

Throughout the whole experiment, the expression of the ECM was basically the same as the WNT signal pathway and HIFs. The expression of ECM was enhanced when the level of the WNT signal pathway and HIFs increased, which was consistent with the research of Pfander et al. [52]. The WNT signal has been identified as one of the major signal pathways to regulate chondrogenesis. As β-catenin could promote chondrocyte hypertrophy and collagen-X was mainly produced by hypertrophic chondrocytes, the high-level of β-catenin and collagen-X in group H showed that MP might stimulate hypertrophy of chondrocytes, which might be connected with serious FHN as well. Moreover, the level of HIFs was positively associated with catabolic factors. It was consistent with the research of Yudoh et al., in which chondrocytes were treated with IL-1β [36]. Furthermore, HIF-1 could regulate chondrocyte proliferation and improve the expression of anti-apoptosis genes [36], it is also in accordance with our results. Therefore, serum catabolic factors might promote the WNT signal pathway and HIFs, and then enhance the expression of ECM genes. As the VEGF was the target gene of HIFs, the high-level HIFs could promote VEGF to restore ECM homeostasis. The research of Derfoul et al. also demonstrated that glucocorticoid could promote the expression of ECM genes [53].

In summary, both the long-term and impulse MP treatment showed that homeostasis-related cytokines had a visible impact on bone formation and resorption, then affected the bone quality. Furthermore, an impulse MP treatment caused more broilers to have serious FHN, whereas the long-term MP administration caused a more evident change in serum cytokines and had a more obvious inhibition of bone growth. It also implied that MP might destroy articular homeostasis to cause broiler FHN.

## 4. Materials and Methods

### 4.1. Animal Treatment and Sample Collection

All animal work was carried out in accordance with the Guidelines for Laboratory Animals of the Ministry of Science and Technology (2006, Beijing, China), and the agreement was approved by the Animal Protection and Use Committee of Nanjing Agricultural University (#NJAU-Poult-2019031804, approved on March 18, 2019). One-day-old broiler chickens (*Gallus gallus*, AA broilers) were randomly divided into three groups (16 chickens per group) and all birds were reared in a standardized process and received basal diet (see Table 8). When the broilers were eleven days old, they were vaccinated according to the procedures for Newcastle Disease, Infectious Bronchitis Disease, Avian Influenza (Subtype H9), and Infectious Bursal Disease. At the age of 29 d, the two experimental groups were treated with different methods of administration: Group L, broilers were injected intramuscularly with MP twice a week for four weeks (12.5 mg·kg^−1^, Haisco Pharmaceutical Group, Liaoning, China); group H, the broilers were injected intramuscularly with MP (20 mg·kg^−1^·d^−1^) for 7 d (pulse treatment). The birds in group C were treated with an isodose sterile saline as the control group. During the whole breeding period, changes in the behavior, BW, and food consumption of broilers in three groups were recorded. The ability of the bird to walk was scored on a six-point scale [54] (see Table 1). Feed conversion (kg of feed per kg of obtained BW) was calculated according to Sakthivelan et al. [55]. The birds in each group were sacrificed at 42 and 56 d.

Blood samples were collected from the jugular vein, each serum was divided into two copies and stored at −20 °C. The liver was collected and the weight was recorded. Bone samples, including femur, tibia, and humerus were collected and cleaned of all adherent tissue. The femoral head was cut along the sagittal plane, carefully washed with physiological saline. One half was cleaned with PBS (Beijing Solarbio Science & Technology Co., Ltd., Beijing, China) and fixed in 4% paraformaldehyde at 4 °C, and the other half was treated with DEPC and then stored in liquid nitrogen.

### 4.2. ELISA and Biochemical Analysis

The frozen serum was transferred at a melting ice bath temperature. The indicators were measured by the chicken specific ELISA kit (Nanjing Angle Gene Biotechnology Co. Ltd., Nanjing, China), each sample was repeated three times. The kits were used to detect two kinds of indicants: Three indicants related to cartilage homeostasis, containing IL-1β, IGF-1, and vascular endothelial growth factor (VEGF); and four bone-related indicants of chicken containing BALP, OT, TRACP-5b, and CTX-I.

The levels of serum creatinine, ALP, Ca, and P were detected by an automatic biochemical analyzer (Hitachi Ltd., Tokyo, Japan), each sample was repeated three times.

### 4.3. Histopathological Analysis

After washing overnight at room temperature, the fixed cartilage tissues were decalcified in 10% EDTA for two weeks. After dehydration with ethanol, hyalinization with dimethylbenzene, and embedment in paraffin, the fragments were cut into 4 μm-thick pieces and stained with hematoxylin and eosin (H&E) for pathological observation.

### 4.4. Bone Parameters Detection

The BMD of the left femur, tibia, and humerus were measured by a dual-energy X-ray absorption measuring instrument (Medikors, Inc., Gyeonggi-do, Korea). The test mode was set to fast scan with a high energy parameter of 80 kVp/1.0 mA and a low energy parameter of 55 kVp/1.25 mA. The acquired images were analyzed using the InAlyzer 1.0 image processing system. Then, the length, index, and diameter of the midpoint of these bones were measured and recorded.

### 4.5. RNA Extraction and Real-Time Quantitative PCR

The femoral head was grinded into powder in a liquid nitrogen (−196 °C) environment and treated with Trizol (Nanjing Angle Gene Biotechnology Co. Ltd., Nanjing, China) to extract the total RNA. The complementary DNA (cDNA) was synthesized via reverse transcription utilizing HiScript II QRT SuperMix for qPCR (+gDNA wiper; Zazyme, Nanjing, China). The expressions of the homeostasis-related gene were detected by a quantitative real-time PCR (qRT-PCR) on the ABI PRISM 7300 HT sequence-detection system (Applied Biosystems, Inc., Foster City, CA, USA), repeated three times. The genes selected were collagen-2, aggrecan, β-catenin, LRP5, LRP6, HIF-1α, VEGFA, and IL-1β. Quantitative data were normalized relative to the housekeeping GAPDH. The genes’ primer sequences as described above are listed in Appendix A. All PCR operations were performed in triplicate. The results were analyzed as the relative fold change (2^−ΔΔCT^ value) [56].

### 4.6. Statistical Analysis

A statistical analysis was conducted using IBM SPSS Statistics 19. The morbidity of FHN in different groups was evaluated by X^2^ tests. All other data were presented as the mean ± SE and the differences between groups were determined with one-way analysis of variance (ANOVA, LSD). Significant differences were accepted if *p* < 0.05.

## 5. Conclusions

In conclusion, both the long-term MP injection (12.5 mg·kg^−1^, twice a week for four weeks) and impulse MP injection (20 mg·kg^−1^·d^−1^ for seven consecutive days) can raise the morbidity of FHN in broilers. The impulse MP treatment is recommended due to its short modeling period (from 29 to 42 d) and high incidence of serious FHN (femoral head separation with laceration of the growth plate). This study also provides a reference for the clinical use of MP in birds. Injecting intramuscularly with MP at both doses of 20 and 12.5 mg·kg^−1^ may cause broiler death, and broilers have diarrhea during continuous injection at the dose of 20 mg·kg^−1^·d^−1^. In addition, the MP treatment inhibited the growth of bones and caused bone mineral loss, which may induce osteoporosis. The results also implied that MP might destroy articular homeostasis to cause broiler FHN. Due to the destroyed homeostasis of cells and ECM, the mRNA expressions of the WNT signal pathway and HIFs could be promoted and then increase the transcription level of ECM, which helps synthesize ECM and restore homeostasis.

## Figures and Tables

**Figure 1 ijms-21-04841-f001:**
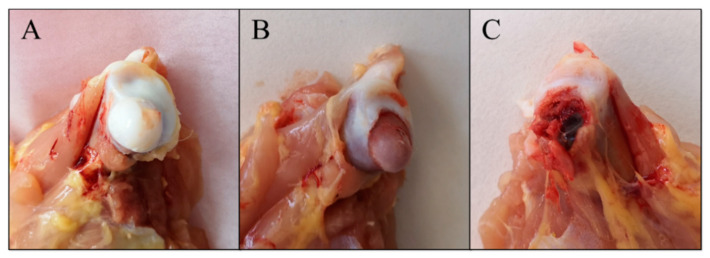
The macroscopic morphological changes for different degrees of femoral head necrosis (FHN). (**A**) The normal femoral head with an integral structure. The FHN evaluation score is set to 0. (**B**) The femoral head separates from the growth plate. The FHN evaluation score is set to 1. (**C**) Rupture of the femoral head. The FHN evaluation score is set to 2.

**Figure 2 ijms-21-04841-f002:**
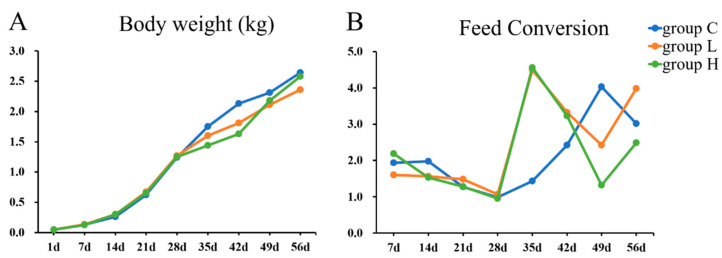
(**A**) The change of body weight from 1 to 56 d. (**B**) The trend of feed conversion at each week from 7 to 56 d.

**Figure 3 ijms-21-04841-f003:**
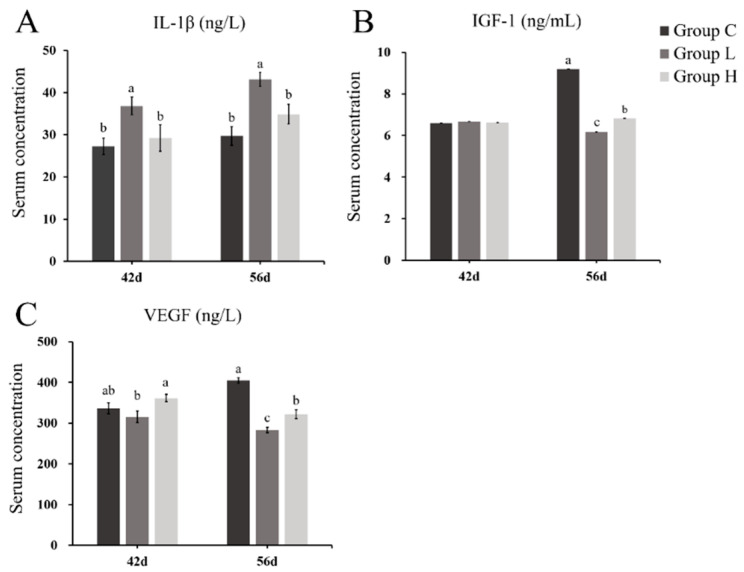
The changes of serum indicants related to extracellular matrix (ECM) homeostasis. (**A**) The changes of serum catabolic cytokines interleukin-1β (IL-1β) in different groups. (**B**,**C**) The changes of serum anabolic cytokines insulin-like growth factor 1 (IGF-1) and vascular endothelial growth factor (VEGF) in different groups. Superscripts with different letters indicate significant differences (*p* < 0.05).

**Figure 4 ijms-21-04841-f004:**
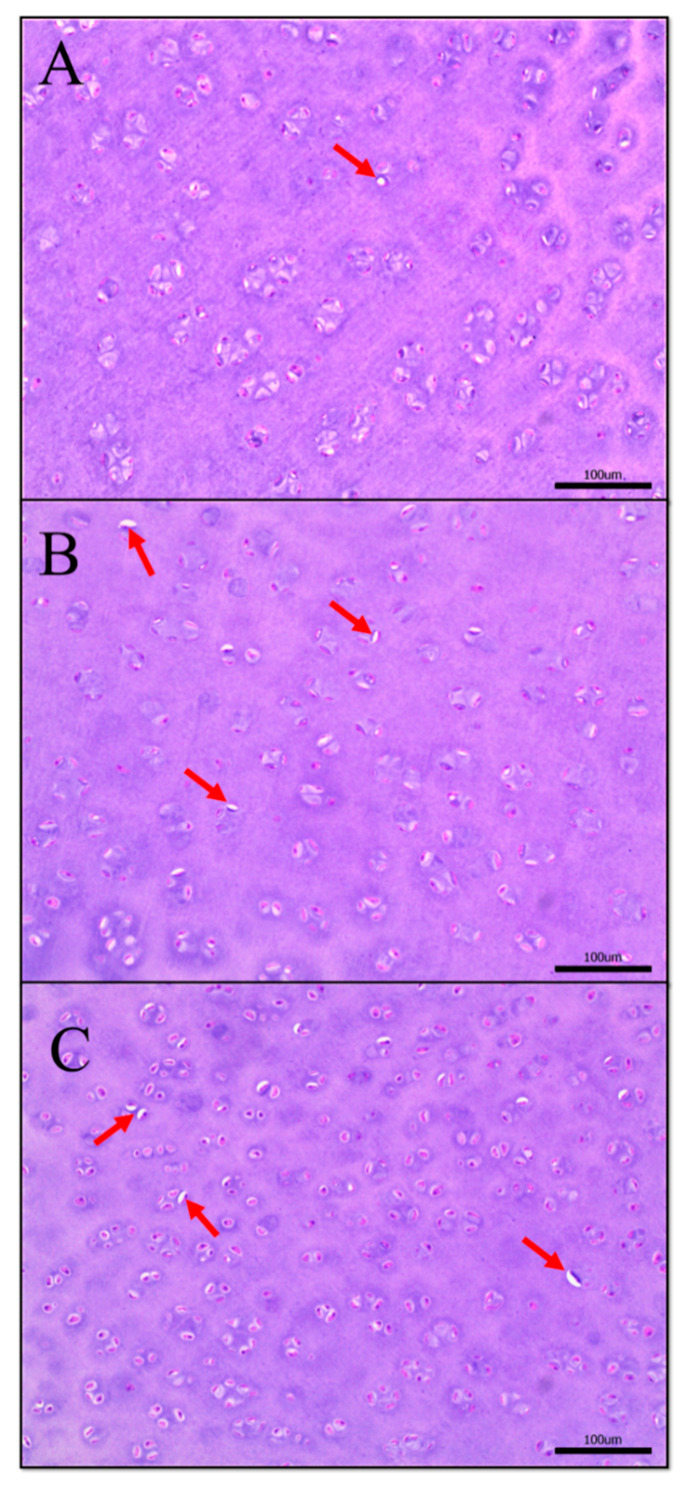
Hematoxylin and eosin (H&E) staining of chondrocytes in the femoral head articular cartilage (scale bar: 100 μm). (**A**) H&E stained histological section of the normal femoral head in group C. Histological section of the femoral head cartilage in group L (**B**) and group H (**C**). The area where the red arrow points is an empty lacunae.

**Figure 5 ijms-21-04841-f005:**
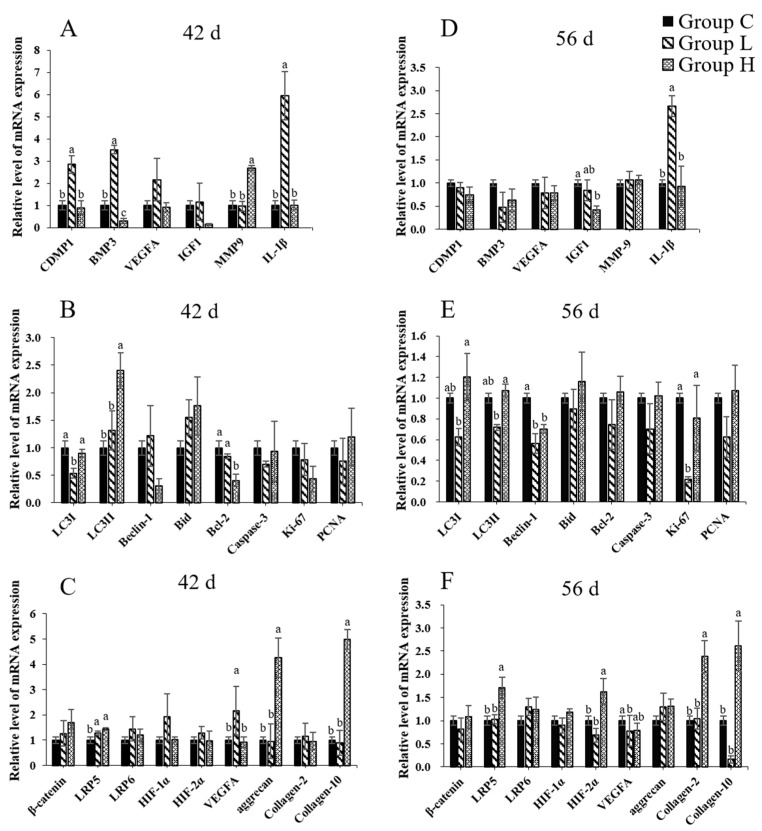
The mRNA expression of chondrocytes in three groups. (**A**) The 42-d expression of ECM-homeostasis relative genes. (**B**) The 42-d expression of cell-homeostasis relative genes. (**C**) The 42-d expression of the WNT signal pathway and hypoxia-inducible factors (HIFs). (**D**) The 56-d expression of ECM-homeostasis relative genes. (**E**) The 56-d expression of cell-homeostasis relative genes. (**F**) The 56-d expression of the WNT signal pathway and HIFs. Superscripts with different letters indicate significant differences (*p* < 0.05).

**Table 1 ijms-21-04841-t001:** The numbers of broilers with each gait score.

Gait Scoring (0–5)	42 Day			56 Day		
	Group C (*n* = 14)	Group L (*n* = 14)	Group H (*n* = 14)	Group C (*n* = 8)	Group L (*n* = 8)	Group H (*n* = 8)
0	2	1	0	1	0	0
1	5	3	2	2	1	0
2	6	6	5	4	3	2
3	1	4	5	1	3	4
4	0	0	2	0	1	1
5	0	0	0	0	0	1

Note: Gait score 0: The bird walked normally with no detectable abnormality; Gait score 1: The bird had a slight defect which was difficult to define precisely; Gait score 2: The bird had a definite and identifiable defect in its gait; Gait score 3: The bird had an obvious gait defect which affected its ability to move about; Gait score 4: The bird had a severe gait defect; Gait score 5: The bird was incapable of sustained walking on its feet.

**Table 2 ijms-21-04841-t002:** Femoral head necrosis (FHN) evaluation and morbidity in broilers, both legs of each broiler were counted.

Item		42 Day			56 Day		
		Group C (*n* = 6)	Group L (*n* = 6)	Group H (*n* = 6)	Group C (*n* = 8)	Group L (*n* = 8)	Group H (*n* = 8)
FHN evaluation score (0–2)	0	8	9	7	12	9	11
	1	3	3	2	4	7	2
	2	1	0	3	0	0	3
Morbidity Score 1 to 2 (%)		33.33	25.00	41.67	25.00	43.75	31.25
Total disease score		5	3	8	4	7	8
Amount		12	12	12	16	16	16

Note: FHN evaluation score 0: The normal femoral head with an integral structure. FHN evaluation score 1: The femoral head separates from the growth plate. FHN evaluation score 2: Femoral head separation with laceration of the growth plate.

**Table 3 ijms-21-04841-t003:** The changes of body weight, liver weight, and liver index in broilers.

Item		Group C	Group L	Group H
Body weight (kg)	42 day	2.13 ± 0.08 ^a^	1.81 ± 0.12 ^b^	1.63 ± 0.14 ^b^
	56 day	2.64 ± 0.12 ^a^	2.36 ± 0.09 ^b^	2.58 ± 0.10 ^a^
Liver weight (g)	42 day	44.66 ± 2.04 ^a^	35.61 ± 3.15^b^	35.53 ± 2.42 ^b^
	56 day	36.80 ± 1.51	36.02 ± 0.90	37.79 ± 2.74
Liver index (%)	42 day	20.03 ± 0.51	19.59 ± 0.43	20.54 ± 1.07
	56 day	14.03 ± 0.47 ^b^	16.03 ± 0.78 ^a^	14.60 ± 0.65 ^ab^

Note: Different superscript letters in the same line means there is a significant difference (*p* < 0.05).

**Table 4 ijms-21-04841-t004:** The changes of serum biochemical indicators in broilers.

Item		Group C	Group L	Group H
Creatinine (umol/L)	42 day	22.83 ± 1.51	20.20 ± 0.80	23.83 ± 2.82
	56 day	21.75 ± 0.86	21.00 ± 0.80	21.56 ± 1.25
ALP (U/L)	42 day	2733.00 ± 215.33	2987.60 ± 862.03	3867.00 ± 687.64
	56 day	3466.88 ± 668.62	3267.13 ± 322.33	5032.33 ± 813.02
Ca (mmol/L)	42 day	2.68 ± 0.05	2.67 ± 0.08	2.68 ± 0.10
	56 day	2.58 ± 0.05 ^b^	2.98 ± 0.08 ^a^	2.58 ± 0.03 ^b^
P (mg/dL)	42 day	2.14 ± 0.09 ^a^	1.82 ± 0.05 ^b^	1.99 ± 0.07 ^ab^
	56 day	2.13 ± 0.06	2.09 ± 0.09	2.17 ± 0.07

Note: Different superscript letters in the same line means there is a significant difference (*p* < 0.05).

**Table 5 ijms-21-04841-t005:** The changes of serum indicants related to bone metabolism in broilers.

Item		Group C	Group L	Group H
BALP (pg/mL)	42 day	304.89 ± 6.51	292.95 ± 16.23	304.64 ± 6.44
	56 day	462.17 ± 15.13 ^a^	230.87 ± 4.55 ^b^	254.99 ± 9.18 ^b^
OT (ug/L)	42 day	18.75 ± 0.41	18.75 ± 0.56	19.55 ± 0.35
	56 day	24.17 ± 0.47 ^a^	18.50 ± 0.83 ^b^	19.79 ± 0.53 ^b^
TRACP5b (ng/L)	42 day	1366.11 ± 6.24 ^b^	1439.23 ± 22.08 ^a^	1312.06 ± 21.70 ^c^
	56 day	916.57 ± 51.62 ^b^	1463.11 ± 52.03 ^a^	1329.64 ± 39.81 ^a^
CTX-I (ng/mL)	42 day	207.16 ± 4.57 ^b^	252.91 ± 13.26 ^a^	212.78 ± 3.16 ^b^
	56 day	186.16 ± 3.50 ^b^	262.17 ± 3.81 ^a^	249.83 ± 13.84 ^a^

Note: Different superscript letters in the same line means there is a significant difference (*p* < 0.05).

**Table 6 ijms-21-04841-t006:** The 42-d changes of bone parameters in broilers.

Group		Density (g/cm^2^)	Length (cm)	Index (g/kg)	Diameter (mm)
Humerus	Group C	0.26 ± 0.04	6.95 ± 0.09 ^a^	2.65 ± 0.05	0.85 ± 0.02 ^a^
	Group L	0.20 ± 0.01	6.19 ± 0.13 ^b^	2.37 ± 0.18	0.73 ± 0.01 ^b^
	Group H	0.23±0.03	6.26 ± 0.14 ^b^	2.53 ± 0.05	0.75 ± 0.03 ^b^
Femur	Group C	0.20 ± 0.01	8.53 ± 0.11 ^a^	5.08 ± 0.14 ^a^	1.02 ± 0.02 ^a^
	Group L	0.20 ± 0.01	7.26 ± 0.16 ^b^	4.60 ±0.15 ^b^	0.89 ± 0.02 ^b^
	Group H	0.19 ± 0.01	7.37 ± 0.19 ^b^	5.13 ± 0.16 ^a^	0.94 ± 0.02 ^b^
Tibia	Group C	0.23 ± 0.01^a^	11.30 ± 0.12 ^a^	7.57 ± 0.16 ^a^	0.95 ± 0.01 ^a^
	Group L	0.22 ± 0.01 ^ab^	9.81 ± 0.11 ^b^	6.78 ± 0.24 ^b^	0.77 ± 0.02 ^b^
	Group H	0.20 ± 0.01 ^b^	9.88 ± 0.29 ^b^	7.42 ± 0.31 ^ab^	0.82 ± 0.02 ^b^

Note: Different superscript letters in the same column (the same kind of bone) means there is a significant difference (*p* < 0.05).

**Table 7 ijms-21-04841-t007:** The 56-d changes of bone parameters in broilers.

Group		Density (g/cm^2^)	Length (cm)	Index (g/kg)	Diameter (mm)
Humerus	Group C	0.23 ± 0.00	7.87 ± 0.18 ^a^	2.70 ± 0.19	0.91 ± 0.02 ^a^
	Group L	0.22 ± 0.00	7.33 ± 0.09 ^b^	2.56 ± 0.11	0.82 ± 0.02 ^b^
	Group H	0.22 ± 0.01	7.41 ± 0.11 ^b^	2.42 ± 0.10	0.84 ± 0.02 ^b^
Femur	Group C	0.20 ± 0.01	9.28 ± 0.20 ^a^	5.97 ± 0.24 ^a^	1.04 ± 0.04 ^b^
	Group L	0.19 ± 0.01	8.39 ± 0.09 ^b^	4.85 ± 0.09 ^b^	0.99 ± 0.02 ^b^
	Group H	0.20 ± 0.00	8.90 ± 0.14 ^a^	5.55 ± 0.12 ^a^	1.11 ± 0.03 ^a^
Tibia	Group C	0.24 ± 0.00	12.53 ± 0.25 ^a^	8.18 ± 0.29 ^a^	1.00 ± 0.03 ^a^
	Group L	0.22 ± 0.01	11.51 ± 0.18 ^b^	7.29 ± 0.13^b^	0.88 ± 0.02 ^b^
	Group H	0.23 ± 0.00	11.90 ± 0.15 ^b^	7.99 ± 0.23 ^a^	0.97 ± 0.02 ^a^

Note: Different superscript letters in the same column (the same kind of bone) means there is a significant difference (*p* < 0.05).

**Table 8 ijms-21-04841-t008:** Ingredient compositions and the nutrient levels of elemental diets (dry matter basis, %).

Ingredient	Starter (from 1 to 21 d)	Grower (from 22 to 56 d)
Corn	57.00	62.00
Soybean meal	32.60	28.00
Corn gluten meal	3.00	2.00
Soybean oil	3.00	4.00
CaHPO4	2.00	1.60
Limestone	1.23	1.30
L-Lysine	0.32	0.31
NaCl	0.30	0.30
DL-Methionine	0.15	0.11
Premix *	0.40	0.38
Total	100.00	100.00
Ca level	1.00	0.93
Available P level	0.46	0.39

Note: * Per kilogram of premix containing 12,000 IU Vitamin A, 3000 IU Vitamin D3, 30 IU Vitamin E, 1.3 mg Vitamin K3, 0.013 mg Vitamin B12, 400 mg choline chloride, 40 mg niacin, 10 mg calcium pantothenate, 8 mg riboflavin, 4 mg pyridoxine, 2.2 mg thiamine, 1 mg folic acid, 0.04 mg biotin, 110 mg manganese, 80 mg iron, 65 mg Zinc, 7.5 mg copper, 1.1 mg iodine, and 0.3 mg selenium.

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
