# Peer review of "Cartilage Homeostasis Affects Femoral Head Necrosis Induced by Methylprednisolone in Broilers"

_ijms, 2020, doi:10.3390/ijms21144841_

Round 1

Reviewer 1 Report

Cartilage Homeostasis Affects Femoral Head Necrosis Induced by Methylprednisolone in Broilers

Major concerns

  1. Grammars and Writing should be corrected and re-written thoroughly and intensively.

Ex. reports indicated…. (line 11)

Ex. three groups (n=16/group); group L with methylprednisolone (MP) injection through intramuscular route (12.5 mg•kg-1) twice a week for four weeks; group H  with intramuscular injection of MP (15 mg·kg-1·d-1) for 7 d (pulse treatment); and control group C with with sterile saline. (line 13-15)

  1. Moreover, impulse MP injection increased the portion of birds with severer FHN….. MP injection .. in serum cytokine concentrations and bone metabolism indicators (line 20-22)
  2. FHN broilers induced by MP showed more pronounced production of catabolic factors and suppressed the anabolic factors…(line 23-24).
  3. commercial poultry industry …a significant rapid ..(line 33)
  4. The authors need to specify the ELISA kits used in the study, chicken specific? if not, the authors need to show evidences for validation. Also, in TNF-a analysis, so far no solid evidences showing real TNF-a gene in chicken genome despite that Western and ELISA studies showed positive signals. So, the results of TNF-a analysis are suggested to be deleted.
  5. The authors need to state more evidences that the prevalence of FHN in modern broiler chickens as compared to the past. Also, since modern broilers are created by genetic selection for rapid early growth along the time, the etiology of FHN is involved with a variety of gene expression changes by selection even though the pathogenic progression is similar to pharmacologically-induced FHN such as MP.
  6. There are several similarities between Tibial dyschondroplasia (TD) and FHN. The authors may discuss their relationship. Did the authors examine TD in brirds with MP injection?

Reviewer 2 Report

Manuscript revision ID: ijms-844072 titled:

Cartilage homeostasis affects femoral head necrosis induced by methylprednisolone in broilers

Abstract

Please explain the abbreviations that appear for the first time in the text. Complete the methodological part, it is very poor, e.g. did you use a control group? There are no recommendations for veterinarians for use in poultry farming.

L72 – specify the aim of the study, please. What catabolic and anabolic changes have been studied? What about mineral elements?

L 76 – incorrect arrangement of the manuscript. There should be a material and method chapter here.

How many birds were examined, how many were slaughtered? How big were the birds? What nutrition was used? What was the composition of the mixture: component and chemical? Have any feed additives been used? Preservatives, probiotics, coccidiostats, etc. ?? This can significantly affect the results of observations !! This information is absolutely necessary! What was the content of mineral elements in it? How many replicates of sampling, measuring and analyzing were there? In what group were behavioral studies conducted? What was the percentage of extremely behaving birds?

Tab. 1 no description of the groups. How was FC determined? On what group of birds?

L 103 - Ca? Was marked? What level was in the feed mixtures?

Discussion -please correct it

L183-199 - no direct reference to the data presented

L226-242 - discussion weakly related to the presented research

Conclusion

What are the recommendations for practice? What is recommended for wet doctors?

Round 2

Reviewer 2 Report

In Abstract, the authors maintained the correct arrangement of content. Why is it not maintained throughout the manuscript? The Material and Methods chapter should be No. 2. The paper should not be published without this correction. The content pattern in the manuscript does not meet the standards of scientific publishing

L84 What about mineral elements? The authors did not respond to the question?

L 85 – invalid manuscript structure. There should be a material and method chapter here. - The authors did not respond to my attention again. I see that the chapter Material and methods is after Discussion. This is not the answer I expected, nor the correction made by the authors.

Table 1  feed mixtures for chickens 1-21d of age are called - starter, and from 21-56 days of age - grower / finisher

Table 2 and 3 – keep the scheme as in Table 4, 5, 6…

Table 5 – what about P? What was the P level available in the feed mixtures? If its blood level was tested, it should also be included in the feed as the starting level. There is no description in the chapter Material and methods on the method of determining mineral element

Conclusion

What are the recommendations for practice? What is recommended for wet doctors? Even from model analysis - the authors did not refer to it ...
